# Comorbidity of mental ill-health in tuberculosis patients under treatment in a rural province of South Africa: a cross-sectional survey

Yanga Thungana ![ORCID],[1,2] Robert Wilkinson,[3,4,5] Zukiswa Zingela[6]

For numbered affiliations see end of article.

**Correspondence to**
Dr Yanga Thungana;
ythungana@wsu.ac.za

## ABSTRACT

**Objectives** Tuberculosis (TB) remains prevalent despite the availability of effective anti-TB medications, and accumulating evidence suggests a high rate of mental disorders in people with TB. This is because TB and psychiatric disorders share several risk factors, such as poverty, homelessness and substance use disorder. Moreover, psychiatric comorbidities in patients with TB are associated with poor treatment outcomes. This study explored the psychiatric comorbidity and clinical correlates in individuals receiving TB treatment.

**Design** A cross-sectional survey over 10 months.

**Setting** Two primary care clinics at King Sabata Dalindyebo district, Mthatha, Eastern Cape, South Africa.

**Participant** Patients receiving TB treatment in the two clinics.

**Intervention** The Mini-International Neuropsychiatric Interview was used to screen for psychiatric disorders.

**Primary and secondary outcome measures** Rates of mental disorders in patients with TB over a 10-month period. Variation in rates by sex, employment status and HIV comorbidity.

**Results** In a sample of 197 participants, most patients were men (62%) and screened positive for a mental disorder (82%) with anxiety (48%), depression (38%) and substance use disorders (43%) being the most common psychiatric conditions. On average, individuals had 4 (SD 2) mental disorders. Females had higher rates of depression (p=0.005) and non-adherence to TB treatment (p=0.003), and alcohol use disorder was more common in males (p<0.001) and in those non-adherent to TB treatment. Additionally, low education levels and unemployment were associated with depressive and anxiety disorders (p<0.05).

**Conclusions** Mental disorders are common in patients with TB, and mental health services need to be integrated into the management of patients with TB. Factors linked to mental disorders in this cohort, such as low education, gender and unemployment, may be useful for compiling a risk profile to help identify those with TB who may require more intensive support for their mental health.

## INTRODUCTION

Despite the availability of effective anti-tuberculosis (anti-TB) drugs, TB remains one of the top 10 causes of death worldwide,

## STRENGTHS AND LIMITATIONS OF THIS STUDY

⇒ Using a structured clinical diagnostic interview improved consistency in the individual assessments.
⇒ The Mini-International Neuropsychiatric Interview covers a broad range of mental disorders, and the most common psychiatric disorders are included in the scale.
⇒ As a cross-sectional study, causal inferences could not be drawn between variables.
⇒ The inclusion of only rural participants and a small sample could limit the generalisability of study findings.

and it is among the leading causes of death from a single infectious agent.[1] South Africa is among the top eight countries that account for two-thirds of the global TB infections.[1] Additionally, in people infected with TB, there are high rates of mental disorders such as depression and anxiety disorders.[2–5] Depression is one of the most common comorbid psychiatric disorders in people with TB and is estimated to be three times higher than in patients without TB.[6–8] Conversely, people with psychiatric disorders may be at an increased risk of developing TB as well.[9–11]

There are several postulated reasons for the increased prevalence of mental disorders in people with TB, including shared risk factors such as homelessness, HIV/AIDS, substance use, stigma, malnutrition and poor socioeconomic status.[12–15] Psychiatric comorbidity in TB is associated with poor treatment outcomes, disability, treatment failure and poor quality of life due to poor health-seeking behaviours and challenges with adherence to medication in this cohort of patients.[16–18] Substance use, especially alcohol use, is prevalent in patients with TB and is thought to increase TB incidence, TB mortality, delayed presentation to health facilities and poor adherence to TB treatment.[19–21] Furthermore, there is evidence

that treating comorbid psychiatric disorders can improve TB outcomes.[22]

Although the relationship between TB and mental disorders is described, there is often a lack of integration of mental health services in the management of patients with TB.[23] Additionally, in South Africa, limited studies have explored in detail the prevalence of psychiatric disorders in people with TB. The value of this study is in the use of a different psychiatry questionnaire (Mini-International Neuropsychiatric Interview (MINI)) that is more extensive and evaluates many more different psychiatric disorders compared with previous studies evaluating mental disorders in patients with TB. Also, unique to our study is that the MINI is observer administered compared with the commonly used self-report questionnaires. Therefore, we sought to investigate the point prevalence of comorbid mental disorders and its clinical correlates in patients receiving TB treatment in two district clinics in a rural area in the Eastern Cape, South Africa.

## METHODS
### Sample and procedure
A cross-sectional study was conducted among patients with TB attending two primary care clinics (Mthatha Gateway and Stanford Terrace) at King Sabata Dalindyebo district, Mthatha, Eastern Cape, South Africa. Both clinics offer general medical care, including TB screening and the management of patients with TB. The study was conducted over a 10-month period, from September 2020 to June 2021. Participants included in the study were all patients currently diagnosed and receiving TB treatment who attended the two clinics during the study period. In these clinics, each morning, a nurse holds a short meeting with patients attending the clinic on the day to triage and guide them on accessing services. It is in these morning meetings that the nurse would identify all patients coming to collect TB treatment and would inform the research assistant (RA), who then approached each patient individually. Due to the COVID-19 pandemic and applicable South African lockdown restrictions at the time, the total number of patients attending clinics was significantly reduced.

The RA would explain the study to each patient individually in their language, obtain written informed consent and assure them that their personally identifying information would be kept confidential at all times. Excluded from the study were participants under 18 years of age and those who could not give informed consent. A total of 197 participants were approached by the RA; all provided written consent and thus were included in the study. For our study, a mental disorder, also referred to as a psychiatric disorder, is a condition with a clinically significant disturbance of an individual's emotions, cognition or behaviour associated with personal distress or impairment in adaptive function. And would include all the conditions recognised in the Diagnostic and Statistical Manual of Mental Disorders, fifth edition (DSM-5).

Psychiatric comorbidity or comorbid psychiatric disorder in our study refers to diagnosing a mental disorder in a patient with current TB.

### Data collection
The RA had prior experience in mental health research and was trained on using the MINI over 10 working days by a psychiatrist. The training was held for 90 min each day regarding how to use the different modules of the MINI and an explanation of the various psychiatric terms used in the MINI. The last two sessions were role-playing interviews to assess the RA's level of understanding of the MINI. Lastly, the RA received supervision throughout the period of data collection from the lead author, who is a qualified specialist psychiatrist and a neuropsychiatrist fellow at the time of the study.

The English version of the MINI was administered in English to all study participants. The RA was instructed to read the questions slowly, repeat them as necessary and encourage the patients to seek clarity on any question that is not entirely clear to them. The RA was instructed to schedule at least an hour to accommodate those slow in answering the numerous questions of the MINI. Lastly, when a patient did not understand a question, the RA was allowed to explain it in the patient's primary language.

### Measures
A data capture sheet was used to collect sociodemographic details, past psychiatric history, medical history including HIV status and adherence to TB treatment. Data on whether a participant had defaulted on their current TB treatment were also sought from the patient, although their responses were not verified through other means. The patients were asked whether they had stopped taking the TB treatment for at least seven consecutive days.

The MINI was developed as a brief, structured diagnostic interview used as a measure to diagnose several psychiatric disorders.[24] It has demonstrated good psychometric properties as a tool to screen for several psychiatric disorders.[24–26] For instance, the specificity of the MINI is at least good to excellent (0.72–1.0) for the diverse conditions screened is used to evaluate.[25 27 28] Although the MINI contains over 100 questions and screens for 17 mental disorders, it can be administered over a short period—approximately 20–30 min. The MINI is organised into diagnostic modules. For nearly all the modules, two to four screening questions are used to exclude a diagnosis if answered negatively. Conversely, positive responses to the screening questions lead to further inquiry about other diagnostic criteria. The MINI modules include a major depressive disorder (current and recurrent), dysthymia, manic and hypomanic episodes (current and past), panic disorder (current and lifetime), agoraphobia, social phobia, obsessive–compulsive disorder (OCD), post-traumatic stress disorder, alcohol abuse and dependence, non-alcohol substance use and dependence, psychotic disorders (current and lifetime), anorexia nervosa, bulimia

nervosa, generalised anxiety disorder and antisocial personality disorder. We used all the MINI modules in our study.

## Patient and public involvement

There was no involvement of patients in the design, recruitment and conduct of the study. The study results will not be distributed to the individual participants, but the published paper will be available in the participating clinics.

## Data analytical plan

Data were analysed using STATA/SE (V.16.1 for Mac). Descriptive statistics such as frequencies and proportions for categorical variables, median and IQR, or mean and SD for continuous variables depending on the data skewness were used to characterise the sample. Inferential statistics were applied to evaluate the association between different variables. First, to assess associations between categorical variables, we performed either the Pearson's $\chi^2$ test or Fischer's exact test, depending on the number of scored items under each variable. Second, for the continuous variables, we first assessed for normality using the Skewness and Kurtosis test with a $p>0.05$ indicating normal distribution. Therefore, due to the skewness of our data, we used either the Wilcoxon rank-sum test or the Kruskal-Wallis test to evaluate associations between numerical and categorical variables. The significance level was set at $p<0.05$ for all the bivariate hypothesis testing.

## RESULTS

Most of the 197 participants were males (61.9%, n=122), and the median age was 34 years (IQR 28–42) of age (table 1). Most of the participants were single (70.6%, n=139), had at least secondary education (56.9%, n=112) and were unemployed (57.4%, n=113). We did not find any statistically significant associations between age and the different psychiatric disorders. For instance, the adult group (44%, n=66) had the highest proportion of participants with an alcohol use disorder compared with other age groups, but it was not statistically significant (p=0.055). Unemployed participants were more likely to have an anxiety disorder (p=0.03), depressive disorder (p<0.001) and OCD (p=0.006). Moreover, individuals earning less than R2500.00 a month had a higher proportion of depressive disorders (74.7%, p<0.001), social phobia (75.4%, p=0.03) and agoraphobia (85.7%, p=0.01) than those earning more than R2500.00. Lower levels of education were associated with recurrent depressive disorder (p=0.001), lifetime panic disorder (p=0.02), ongoing generalised anxiety disorder (p=0.02) and current OCD (p=0.03). But social anxiety disorder was more common in those with at least a high school education (p=0.02).

**Table 1** Sociodemographic details of the participants (N=197)

| | Number | Percentage |
|---|---|---|
| **Sex** | | |
| Male | 122 | 61.9 |
| Female | 75 | 38.1 |
| **Age** | | |
| Adolescents (<19 years) | 4 | 2 |
| Adults (19–44) | 149 | 76 |
| Middle age (45–64) | 33 | 16.8 |
| Aged+ (65+) | 10 | 5.1 |
| **Relationship status** | | |
| Single | 139 | 70.6 |
| Married | 44 | 22.3 |
| Divorced/separated | 14 | 7.1 |
| **Level of education** | | |
| Primary school | 33 | 16.8 |
| High school | 112 | 56.9 |
| Tertiary education | 52 | 26.4 |
| **Employment status** | | |
| Employed | 84 | 42.6 |
| Unemployed | 113 | 57.4 |
| Receiving state grant/pension | 26 | 13.2 |
| **Monthly income*** | | |
| Less than R2500 | 127 | 64.5 |
| More than R2500 | 70 | 35.5 |
| **HIV status** | | |
| Negative | 62 | 31.5 |
| Positive | 128 | 64.97 |
| Unknown | 7 | 3.6 |
| **Taking ART** | | |
| Yes | 99 | 77.3 |
| No | 29 | 22.6 |
| **Disease classification** | | |
| Pulmonary | 153 | 77.7 |
| Extrapulmonary | 44 | 22.3 |
| **Phase of TB treatment** | | |
| Active | 102 | 51.8 |
| Continuation | 95 | 48.2 |

*South Africa poverty line is R1220.
ART, antiretroviral therapy; TB, tuberculosis.

## Medical history

The majority of the participants were HIV positive (65%, n=128) (95% CI 58.0 to 71.4) with significantly higher proportion of males participants (78.4%, n=58) having HIV than females participants (60.7, n=71) (p=0.01). Most participants with an HIV diagnosis were receiving antiretroviral therapy (77.3%, n=99) (table 1). Most

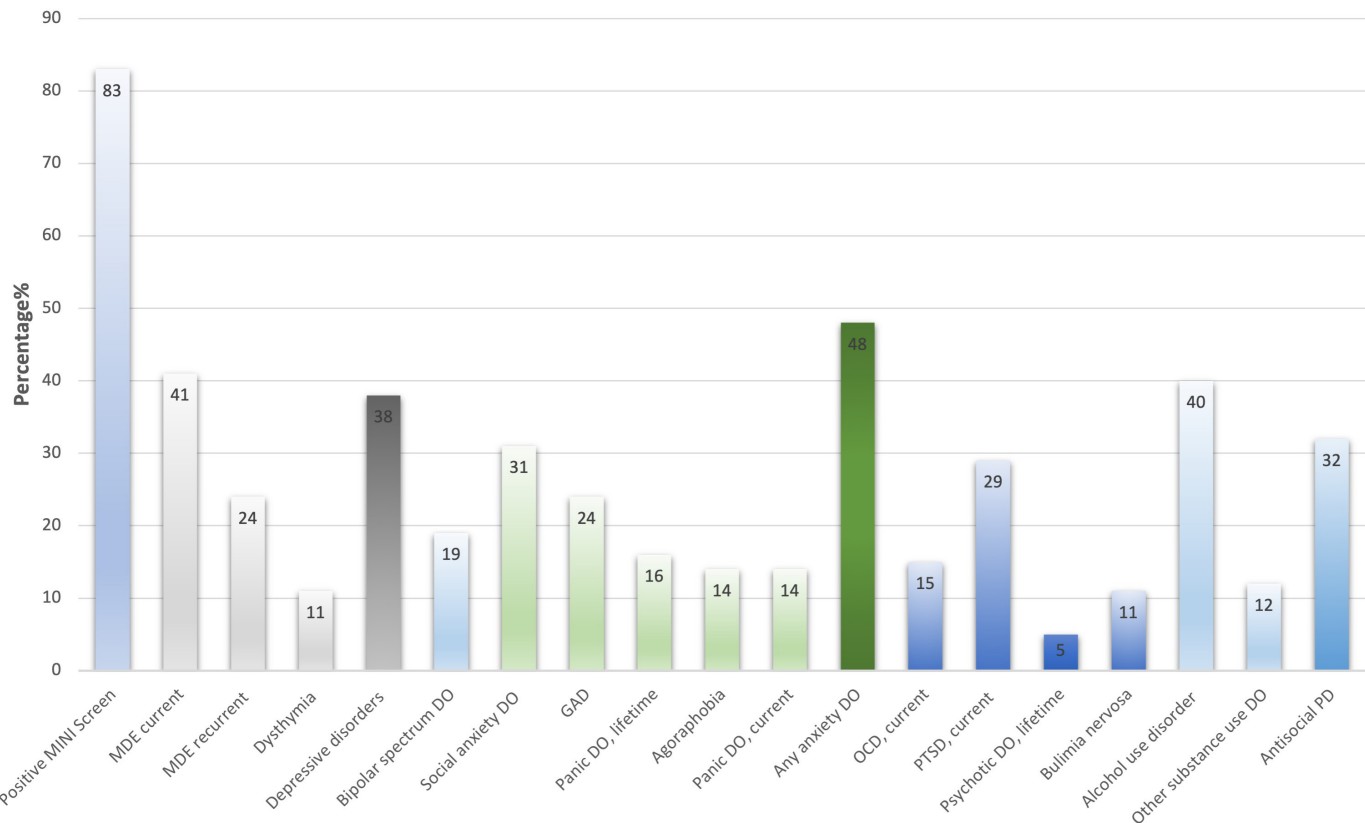

**Figure 1** A proportion of patients with mental disorders among tuberculosis patients (N=197). DO, disorder; GAD, generalised anxiety disorder; MDE, major depressive disorder; MINI, Mini-International Neuropsychiatric Interview; OCD, obsessive-compulsive disorder; PD, personality disorder; PTSD, post-traumatic stress disorder.

HIV-positive (96.6%, n=28) participants not on ART were still in the intensive phase of TB treatment. Moreover, most of the participants in the sample (77.7%, n=153) had pulmonary TB, and more than half (51.8%, n=102) were in the intensive phase of TB treatment. Only a minority of patients (9.1%, n=18) reported non-adherence to their TB treatment. Most participants reporting non-adherence to TB treatment were HIV positive (77.8%, p=0.4), still in the intensive phase (55.6%, p=0.8) of TB treatment, unemployed (55.6%, p=0.8) and assessed to have a mental disorder (88.9%, p=0.4) but these did not reach statistical significance. However, being female (p=0.003) and having an alcohol use disorder (p=0.001) were significantly associated with non-adherence to TB treatment.

### Mental disorders

Most of the participants (82.7%, n=163) screened positive for at least one psychiatric disorder, excluding substance use disorders (figure 1), and the proportion of males participants (82.1%, n=101) with a positive screen was similar to that of female participants (84%, n=63). Furthermore, the HIV status of the participants was not associated with screening outcomes for psychiatric disorders, with those HIV negative almost screening equally positive for a psychiatric disorder as those who were HIV positive (80.7% HIV negative vs 85.2% HIV positive; p=0.4). There was a comparable proportion of individuals

assessed to have a psychiatric diagnosis in those in the intensive phase (82.5%, n=85) and a continuation phase (83.2%, n=79) of TB treatment.

The most common psychiatric disorders were anxiety disorders (48.2%, n=95), current major depressive episodes (40.6%, n=80), unipolar depression (37.6%, n=74), post-traumatic stress disorder (28.9%, n=57), antisocial personality (23.9%, n=47) and bipolar spectrum conditions (19.3%, n=38). Approximately a quarter (25.7%, n=19) of the patients with unipolar depression currently had suicidal ideation. The patients with a positive diagnosis of a mental disorder had an average of 3.5 (SD 2.3) disorders excluding substance use disorders. Also, patients tended to have more than one anxiety disorder with an average of 2 (SD 1) disorders per individual.

More participants (47%) in the intensive phase of TB treatment had a current depressive episode compared with those in the continuation phase (34%); however, this was not statistically significant (p=0.056). Noteworthily, female participants were more likely to have a depressive disorder than male participants (p=0.005). In contrast, men were more likely to screen positive for an antisocial personality disorder (p<0.001).

### Substance use disorders

A considerable number of the participants had a substance use disorder (43.2%, n=85), with more participants having

an alcohol use disorder (40.1%, n=79) than those having a substance use disorder other than alcohol (12.2%, n=24). However, most of the 24 participants (79.2%, n=19) with other substance use disorders also had alcohol use disorder. Nevertheless, both alcohol use disorders (p=0.001) and other substance use disorders (p=0.001) were more common among male participants. In this sample, there was no significant association between HIV status and substance use disorders.

## DISCUSSION

In this study, we examined the psychiatric comorbidity in outpatients receiving TB treatment. We found a high rate of mental disorders in patients with TB; depression, anxiety and substance use disorders were the most common mental disorders. Additionally, those with mental disorders approximately had 4 (SD 2) lifetime psychiatric disorders, excluding substance use disorders. Also, alcohol use disorder was prevalent in our study participants and was significantly associated with the male sex and poor adherence to TB treatment. Female sex was associated with depressive disorders and non-adherence to treatment. Lastly, low levels of education and unemployment were associated with a higher rate of depressive, anxiety and OCDs.

The prevalence of psychiatric disorders in patients with TB varies greatly among different studies from less than 20% to about 80% with a trend of higher rates in lower-middle-income countries (LMIC).[21 29–31] In South Africa, the prevalence of psychiatric comorbidity in patients with TB is similar to other LMIC. For instance, the prevalence of depression is documented to be 64.3%, and PTSD at 29.6%.[6 32] But higher rates of psychological distress (81%) have been reported in South Africa.[6 7] In our study, we found an overall higher rate of mental disorders (~83%), but the rates of PTSD and depression were similar and lower to previous studies, respectively.[6 32] The major variation in prevalence in our study could be due to several factors such as the assessment instrument used, a high HIV comorbidity, the sociodemographic factors, or that it is a true reflection of the high psychiatric comorbidity in this population. For instance, HIV comorbidity in patients with TB is associated with higher mental morbidity.[33] Also, South Africa is a middle-income country with a very high TB burden; the personal socioeconomic consequences of TB infection might be amplified, thus leading to more pronounced mental disorders.[1 33] A case in point documented is that in patients with TB, unemployment, poverty and lower educational background are associated with higher psychiatric comorbidity, as was the case in our study.[6 34 35]

Furthermore, the MINI covers a broader number of conditions than scales used in other studies, and in patients with TB, mental disorders have been found to vary significantly depending on the psychiatric screening scale used.[36] Compared with previous studies that tended to assess broader categories of mental disturbance such as

anxiety or psychological distress, we were able to screen for several individual psychiatric diagnoses. Thus, in addition to describing the presence of mental illness, we were able to assess the number of disorders experienced by each individual and we found a tendency for participants to have more than one psychiatric disorder with an average of approx. Overall, 4 (SD 2) disorders per individual, excluding substance use disorders.

In this study, we did not find a statistically significant relationship between HIV-positive individuals and an increased rate of mental disorders. But the finding of increased psychiatric disorders in people living with HIV (PLWH) has been described.[37–39] However, the rate of mental illness in PLWH varies significantly from as low as below 10% to as high as over 80%.[29 40–43] The wide range in the prevalence could be due to the different study designs, study settings and measurement instruments used in different studies. Moreover, not all studies have found an increased rate of mental illness in PLWH.[44]

We found a slightly higher adherence rate to TB treatment (91%) in the current study. Previous studies have documented adherence rates between 70% and 89%.[45–48] The higher rates of treatment adherence in our study could be due to our methodology, as we solely relied on the patient's report and did investigate it further. Non-adherence to treatment in patients with TB can be due to several factors, including HIV seropositivity, substance use, distance to health facilities, male gender, running out of medication, medication factors, the continuation phase of chemotherapy and stigma.[45–47] Additionally, psychiatric comorbidity has been described as another factor with a negative impact on compliance with TB treatment.[5 33 49 50] Also, substance use is associated with poor adherence to TB treatment,[50–52] as was the case in our study. Contrary to findings in the current study, previous studies have found better adherence to TB treatment among female participants.[47 53] Furthermore, the proportion of participants who defaulted TB treatment was comparable between those in the intensive phase and continuation phase of chemotherapy. But the continuation phase of TB treatment is previously associated with a higher rate of non-adherence to TB treatment.[48 54 55]

Our finding that anxiety and depressive disorders are the most common mental disorders in patients with TB is comparable to previous studies.[21 36 56] Moreover, as in our study, women with TB were more likely to suffer from depressive disorders than male patients.[33 36] It has been reported that psychiatric disorders are more common in the earlier phase of TB infection and rates decline as patients recover from TB infection.[21] Our study showed a similar trend with more patients in the intensive phase of treatment having a current major depressive episode than those in the continuation phase; however, the difference did not reach statistical significance.

Alcohol use disorder was common in our study sample, corroborating a finding in previous literature highlighting high rates of alcohol use in patients with TB.[57–60] In those already infected with TB, alcohol abuse is associated with

a lack of cooperation, poor adherence, higher levels of depression and mortality.[19 20] Similar to previous studies, alcohol use disorders were more common among the male participants in our study.[59 61] However, the prevalence of alcohol use disorder (40%) in our study was higher than previously reported and could be because most participants were men, single, unemployed and had lower education levels.[57 60]

## Strengths and limitations of this study

The use of the MINI, a well-established structured diagnostic interview scale that covers a broader number of mental disorders, adds to the study's strength. Furthermore, all patients with TB attending the two clinics were admitted to the study with no exclusion criteria.

The study was conducted during the COVID-19 pandemic, which possibly affected clinic attendance and TB screening and could have contributed to the high prevalence of mental disorders.[62 63] Along with this, the small study sample could restrain the generalisability of the study results to other settings. Additionally, we did not conduct individual clinical assessments to confirm the diagnosis of the MINI, which could possibly increase the positive psychiatric diagnosis. Lastly, the MINI was administered in English to participants to whom English is not their primary language, which might have affected the accuracy of the patient's responses.

## CONCLUSION

Our study showed an increased frequency and number of mental disorders among individuals receiving TB treatment. Moreover, as in previous studies, we found substance use disorders prevalent in patients with TB. Thus, the study outcomes further highlight the need to integrate mental health services in managing patients with TB at the primary care level, including screening all new patients with TB. There is a need for studies to evaluate whether brief interventions for substance use at the primary care level could improve adherence and recovery from TB.

**Author affiliations**
[1]Department of Psychiatry, Nelson Mandela University, Port Elizabeth, South Africa
[2]Department of Psychiatry and Behavioural Sciences, Walter Sisulu University, Mthatha, Eastern Cape, South Africa
[3]Department of Infectious Disease, Imperial College London, London, UK
[4]Pathology, Frans Crick Institute, London, UK
[5]Wellcome Center for Infectious Diseases Research in Africa. Institute of Infect. Disease and Mol. Med and Dept. Med, University of Cape Town, Cape Town, South Africa
[6]Executive Dean's Office, Faculty of Health Sciences, Nelson Mandela University, Port Elizabeth, Eastern Cape, South Africa

**Acknowledgements** We would like to express our gratitude to our research assistant, Simthembile Lindani, who worked tirelessly and completed data collection despite the challenges of the COVID-19 pandemic. We would like to thank Editage (www.editage.com) for English language editing.

**Collaborators** Not applicable.

**Contributors** YT, RW and ZZ were involved in the conception and design of the study. YT analysed the data, and wrote the manuscript. YT, RW and ZZ revised the manuscript. All authors provided their final approval for manuscript submission and agree to be accountable for all aspects of the work. YT takes responsibility for the overall content as the guarantor.

**Funding** This research was Supported by the National Institutes of Health grant number (NIH U19AI115940). RJW is supported by the Francis Crick Institue which receives funding from Wellcome (FC0010218), Cancer Research UK (FC0010218), and the Medical Research Council (FC0010218).

**Competing interests** None declared.

**Patient and public involvement** Patients and/or the public were not involved in the design, or conduct, or reporting, or dissemination plans of this research.

**Patient consent for publication** Not applicable.

**Ethics approval** This study involves human participants. Ethical approval for this study was obtained from the Walter Sisulu University Human Research Committee (protocol number 119/2019) and the South African Department of Health (EC_202006_009). Participants gave informed consent to participate in the study before taking part.

**Provenance and peer review** Not commissioned; externally peer reviewed.

**Data availability statement** Data are available upon reasonable request.

**ORCID iD**
Yanga Thungana http://orcid.org/0000-0001-9854-314X

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
