## [Reviewer comments · BMJ Open]

ARTICLE DETAILS

TITLE (PROVISIONAL)	The Co-Morbidity of Mental Ill-Health in Tuberculosis Patients under Treatment in a Rural Province of South Africa – A cross-sectional survey.
AUTHORS	Thungana, Yanga; Wilkinson, Robert; Zingela, Zukiswa

VERSION 1 – REVIEW

REVIEWER	Loveday, Marian South African Medical Research Council
REVIEW RETURNED	15-Nov-2021

GENERAL COMMENTS	Manuscript: 'Prevalence of Co-Morbid Mental Ill-Health in Tuberculosis Patients under Treatment in a Rural Province of South Africa' (Manuscript ID: bmjopen-2021-058013) Thank you for asking me to review this manuscript which describes the prevalence of mental health disorders in TB patients in a rural area in South Africa. In this cross-sectional study 197 participants were interviewed using the Mini-International Neuropsychiatric Interview (MINI) tool. The MINI is a brief, structured diagnostic interview used as a measure to diagnose several psychiatric disorders which can be administered by appropriate trained clinicians or research assistants. The manuscript is well written and reads easily, although below I have made a couple of grammatical/language comments which will improve the readability of the manuscript. General comments As I understand the study, it was a cross-sectional study and the MINI was administered to each study participant at one timepoint during their TB treatment. The title is therefore misleading, as the study does not measure the prevalence of mental health conditions over 6 months of TB treatment, but at one point during TB treatment. However, the article is just 'too thin' to warrant publication. Both the methods and results sections are together just over 800 words. • In the methods section more detail is needed on the inclusion and exclusion criteria of the study participants. Did participants sign informed consent? Where, how and in what language was the research tool administered? More detail is needed on the MINI. For example, what diagnostic modules are included. Also more specific limitations of the MINI should be included in the limitations section.• In the results section, all that is reported are the sociodemographic and clinical characteristics of the study
--

	participants, together with the proportion of participants with a mental health condition. One of the study objectives is to explore the clinical correlates of mental illness, but all that is reported is HIV status. ART status is not adequately described. Are no other results available? The authors mentioned that data on adherence and treatment outcomes was collected, were any particular mental health disorders associated with poorer adherence or poorer treatment outcomes? Minor grammatical/language comments To increase the readability of the manuscript:  • Reduce the number of transitions throughout the manuscript, that is words like 'additionally', 'furthermore', 'however'. To illustrate my point I have copied lines 24 – 45 from page 4 and pasted them below, removing the transitions. There are several postulated reasons for the increased prevalence of mental illness in people with TB, including shared risk factors such as homelessness, HIV/AIDS, substance use, stigma, malnutrition, and poor socioeconomic status.[12–15] additionally, pPsychiatric comorbidity in TB is associated with poor treatment outcomes, disability, treatment failure, and poor quality of life due to poor health-seeking behaviours and challenges with adherence to medication in this cohort of patients.[16–18] Moreover, sSubstance use, especially alcohol use, is prevalent in patients with tuberculosis and is thought to increase TB incidence, TB mortality, delayed presentation to health facilities, and poor adherence to TB treatment.[19–21] Furthermore, there is evidence that treating comorbid psychiatric disorders can improve TB outcomes.[22]  • Is it necessary to use the word disorder everytime? I have copied and pasted lines 3 – 11 from page 9 below. In two sentences the term disorder is used 6 times. Is it possible to reduce this? Most of the participants (82.7%, n = 163) screened positive for at least one psychiatric disorder, excluding substance use disorders (Fig 1). The most common psychiatric disorders were anxiety disorders (48.2%, n = 95), current major depressive episodes (40.6%, n = 80), unipolar depressive disorders (37.6%, n = 74), post-traumatic stress disorder (28.9%, n = 57),
--	--

REVIEWER	Janse van Rensburg, André University of KwaZulu-Natal
REVIEW RETURNED	08-Mar-2022

GENERAL COMMENTS	Thank you for the opportunity to review this paper, which deals with a well-known and persistent challenge in health systems across the globe. While the study focuses on an important issue, there are several drawbacks that require attention, presented below:
--

	 - The authors refers to a syndemic relationship between TB and mental illness, which is problematic. First, a syndemic has very clear definitions that need to be described based on data – this has been done for TB and depression, but mental illness is incredibly broad and comorbidity would be a better term to use. - It is not entirely clear, given the target journal scope, what the contribution of the paper is to international scholarship. The results of the paper are expected, and well-known and established across multiple studies and contexts. What could another descriptive study add? It might be worth presenting the novelty of the study as drawing from the MINI rather than from the more often used self-report measures? - More information is required on how participants were identified and included in the study, given the potential ethical issues emerging from recruiting patients with stigmatizing conditions from health facilities. - The MINI is usually administered by a health professional, or by allied health professionals who underwent extensive training – a few sentences unpacking the training and mentorship of the interviewer would be helpful. - The issue of language does not seem to come up – was the interview conducted in a local language? Was the MINI translated? And are there any validation concerns regarding measurement of psychiatric constructs in a community where traditional beliefs intertwine with psychiatric understanding among people? - What were the age groups? E.g. what is an adolescent, adult, middle age in the study? - It is not clear why the results of the bivariate analyses were not included? Also, it is not clear why the results do not include bivariate analyses with HIV, income level and age? - The authors claim that the psychiatric morbidity found was high, but provide no standard against which this is claimed. It could be argued that the levels reported are consistent with the population studied, but more comparisons with previous South African studies could have made this more clear. - In the Discussion, low levels of employment and education were associated with high psychiatric comorbidity, but this is related to specific conditions and should be discussed accordingly. - Stating that the importance of the study relates to the potential of alcohol use to lead to contracting TB is confusing since the population is under active TB management.
--	---

VERSION 1 – AUTHOR RESPONSE

Reviewer: 1

Dr. Marian Loveday, South African Medical Research Council

Comments to the Author:

Manuscript: 'Prevalence of Co-Morbid Mental Ill-Health in Tuberculosis Patients under Treatment in a Rural Province of South Africa' (Manuscript ID: bmjopen-2021-058013)

Thank you for asking me to review this manuscript which describes the prevalence of mental health disorders in TB patients in a rural area in South Africa. In this cross-sectional study, 197 participants were interviewed using the Mini-International Neuropsychiatric Interview (MINI) tool. The MINI is a brief, structured diagnostic interview used as a measure to diagnose several psychiatric disorders, which can be administered by appropriately trained clinicians or research assistants.

The manuscript is well written and reads easily, although below I have made a couple of grammatical/language comments which will improve the readability of the manuscript.

Response: We thank the reviewer for acknowledging the merits of our study.

General comments

As I understand the study, it was a cross-sectional study and the MINI was administered to each study participant at one timepoint during their TB treatment. The title is therefore misleading, as the study does not measure the prevalence of mental health conditions over 6 months of TB treatment, but at one point during TB treatment.

Response: We thank the reviewer for pointing this out. We have revised the title of the study as follows:

'Co-Morbidity of Mental Ill-Health in Tuberculosis Patients under Treatment in a Rural Province of South Africa – a cross-sectional survey'

However, the article is just 'too thin' to warrant publication. Both the methods and results sections are together just over 800 words.

- In the methods section more detail is needed on the inclusion and exclusion criteria of the study participants. Did participants sign informed consent? Where, how and in what language was the research tool administered? More detail is needed on the MINI. For example, what diagnostic modules are included. Also more specific limitations of the MINI should be included in the limitations section.

Response: We thank the reviewer and acknowledge the reviewer's suggestion. As a result, we have re-written most of the methodology section. In addition, we have added more information on the sampling procedure, data collection, and how the MINI was used in our study (see highlighted changes on pages 5 and 8).

- In the results section, all that is reported are the sociodemographic and clinical characteristics of the study participants, together with the proportion of participants with a mental health condition. One of the study objectives is to explore the clinical correlates of mental illness, but all that is reported is HIV status. ART status is not adequately described. Are no other results available? The authors mentioned that data on adherence and treatment outcomes was collected, were any particular mental health disorders associated with poorer adherence or poorer treatment outcomes?

Response: We thank the reviewer for raising these points. As a result, we have added more information exploring associations between age and mental disorder diagnosis, HIV status and diagnosis of mental illness, HIV diagnosis and adherence to TB treatment, psychiatric disorders, and non-compliance to TB treatment, and lastly, we evaluated the relationship between the phase of TB treatment and mental illness (changes highlighted in the results section from page 8 to

11). Also, we expanded our discussion section to incorporate the added study findings (pages 14 and 15).

Minor grammatical/language comments

To increase the readability of the manuscript:

- Reduce the number of transitions throughout the manuscript, that is words like 'additionally', 'furthermore', 'however'. To illustrate my point I have copied lines 24 – 45 from page 4 and pasted them below, removing the transitions.

There are several postulated reasons for the increased prevalence of mental illness in people with TB, including shared risk factors such as homelessness, HIV/AIDS, substance use, stigma, malnutrition, and poor socioeconomic status.[12–15] additionally, pPsychiatric comorbidity in TB is associated with poor treatment outcomes, disability, treatment failure, and poor quality of life due to poor health-seeking behaviours and challenges with adherence to medication in this cohort of patients.[16–18] Moreover, sSubstance use, especially alcohol use, is prevalent in patients with tuberculosis and is thought to increase TB incidence, TB mortality, delayed presentation to health facilities, and poor adherence to TB treatment.[19–21] Furthermore, there is evidence that treating comorbid psychiatric disorders can improveTB outcomes.[22]

Response: We acknowledge the reviewer's suggestion, and we will be cognizant of the frequency of use of the transitions in the manuscript.

- Is it necessary to use the word disorder everytime? I have copied and pasted lines 3 – 11 from page 9 below. In two sentences the term disorder is used 6 times. Is it possible to reduce this?

Most of the participants (82.7%, n = 163) screened positive for at least one psychiatric disorder, excluding substance use disorders (Fig 1). The most common psychiatric disorders were anxiety disorders (48.2%, n = 95), current major depressive episodes (40.6%, n = 80), unipolar depressive disorders (37.6%, n = 74), post-traumatic stress disorder (28.9%, n = 57),

Response: We note and acknowledge the reviewer's suggestion. We have revised the section to reduce the use of the word disorder (paragraph 2, page 11).

Reviewer: 2

Dr. André Janse van Rensburg, University of KwaZulu-Natal

Comments to the Author:

Thank you for the opportunity to review this paper, which deals with a well-known and persistent challenge in health systems across the globe. While the study focuses on an important issue, there are several drawbacks that require attention, presented below:

- The authors refer to a syndemic relationship between TB and mental illness, which is problematic. First, a syndemic has very clear definitions that need to be described based on data – this has been done for TB and depression, but mental illness is incredibly broad and comorbidity would be a better term to use.

Response: We thank the reviewer for raising this point and we have removed the word syndemic from the manuscript.

- It is not entirely clear, given the target journal scope, what the contribution of the paper is to international scholarship. The results of the paper are expected, and well-known and established across multiple studies and contexts. What could another descriptive study add? It might be worth presenting the novelty of the study as drawing from the MINI rather than from the more often used self-report measures?

Response: We appreciate the reviewer's comments. We agree with the reviewer and made changes to highlight the unique contribution of our study compared to previous projects (see the last paragraph of the introduction – pages 4 to 5). The MINI evaluates for several individual psychiatric disorders, so, in addition, to identifying the presence of mental illness, it allowed assessment of the number of mental disorders in each person. The finding of many psychiatric diagnoses (average 4 [SD2]) in each person (highlighted in paragraph 2 of page 11 and 2nd paragraph of page 13) further underscores the increased psychiatric comorbidity in TB patients. There aren't many other studies that have been able elaborate on this point as in our study.

- More information is required on how participants were identified and included in the study, given the potential ethical issues emerging from recruiting patients with stigmatizing conditions from health facilities.

Response: We thank the reviewer for raising this point and we have added information on how the patients were identified, approached, and introduced to the study (highlighted changes in the methods section on pages 5 to 6).

- The MINI is usually administered by a health professional, or by allied health professionals who underwent extensive training – a few sentences unpacking the training and mentorship of the interviewer would be helpful.

Response: We acknowledge the reviewer's comment and have included information on the training and supervision the research assistant received during the study (see data collection on page 6).

- The issue of language does not seem to come up – was the interview conducted in a local language? Was the MINI translated? And are there any validation concerns regarding measurement of psychiatric constructs in a community where traditional beliefs intertwine with psychiatric understanding among people?

Response: We thank the reviewer for raising these points. We have added a section explaining how the MINI was administered (highlighted changes on page 6). We used the English version of the MINI and it was administered in English to all the participants. There is a translated Xhosa version that is not validated anywhere, and it has fewer diagnostic modules. However, we did make use of the Xhosa MINI during the training of the research assistant (RA) to familiarize himself with the vocabulary as he was allowed to explain in Xhosa when a participant did not understand a statement. We acknowledge the potential limitation of administering an instrument in English to participants whose primary language is not English. Thus, we have included language as one of the limitations of the study.

- What were the age groups? E.g. what is an adolescent, adult, middle age in the study?

Response: We thank the reviewer for the point of clarity, and we have added the ages that formed the different age groups in Table 1.

- It is not clear why the results of the bivariate analyses were not included? Also, it is not clear why the results do not include bivariate analyses with HIV, income level and age?

Response: We thank the reviewer for pointing this out. We have added more bivariate analysis between age and mental disorder diagnosis, HIV status and diagnosis of mental illness, HIV diagnosis and adherence to TB treatment, psychiatric disorders and non-compliance to TB treatment, income level and psychiatric diagnosis, and lastly, we evaluated the relationship between the phase of TB treatment and mental illness (changes highlighted in the results section from page 8 to 11).

- The authors claim that the psychiatric morbidity found was high, but provide no standard against which this is claimed. It could be argued that the levels reported are consistent with the population studied, but more comparisons with previous South African studies could have made this more clear.

Response: We appreciate the reviewer's comments and have added previous studies that describe the prevalence of mental illness in patients with TB to compare with findings from our study (see changes highlighted on the 2nd paragraph of the discussion section in pages 12 and 13).

- In the Discussion, low levels of employment and education were associated with high psychiatric comorbidity, but this is related to specific conditions and should be discussed accordingly.

Response: We appreciate the reviewer's comments and have amended the discussion to clearly state which disorders were associated with lower levels of education and unemployment (see highlighted last sentence on the first paragraph of the discussion section one on page 12).

- Stating that the importance of the study relates to the potential of alcohol use to lead to contracting TB is confusing since the population is under active TB management.

Response: We recognize the reviewer's comments. The statement meant to reflect that alcohol use is a risk factor for contracting TB, thus, the prevalent alcohol use in the sample could be a consequence of premorbid substance use leading to the individual acquiring TB. We have rephrased the statement (see it highlighted on the paragraph preceding study strengths and limitations in page 15).

VERSION 2 – REVIEW

REVIEWER	Loveday, Marian South African Medical Research Council
REVIEW RETURNED	22-Jul-2022

GENERAL COMMENTS	The Co-Morbidity of Mental Ill-Health in Tuberculosis Patients under Treatment in a Rural Province of South Africa – A cross-sectional survey. Submission ID: bmjopen-2021-058013.R1
---

Thank you for asking me to review the revised version of this manuscript which describes the extent of mental health co-morbidity in TB patients in the Eastern Cape in South Africa. This is a well-written paper, however, I have a number of comments below which need to be addressed.

Consistent use of terminology

I got quite confused reading the paper as the authors used various terms to refer to mental ill-health. These include: mental illness, mental morbidity, mental disorder, psychiatric condition, psychiatric disorder, mental illness, psychiatric comorbidity, comorbid psychiatric disorders, psychiatric diagnoses, mental disturbance. The authors need to decide on one term and use it consistently throughout the manuscript. In addition, this term should be defined in the methodology. If more than one term is needed as the terms describe different things (surely there can't be more than 2 or 3 terms needed at the most), these should all be defined in the methodology.

Results section

It is not necessary to repeat in the text all that is in the table. So for example, from Table 1 only include what you think are the most important variables in the text. The age categories in Table 1 should be clarified – how old are adults or those who are middle aged or aged+?

The results need to be reported in the accepted academic format. For example, when reporting different adherence in HIV-positive vs HIV-negative participants, the accepted format for reporting this is (77.8% HIV positive vs XX% HIV negative; $p=0.4$)

In the abstract, the last sentence in the objectives needs revising as it doesn't make sense: 'This study explored the psychiatric comorbidity and its clinical in individuals receiving TB treatment.'

Page 4, line 5: Use the term integrate rather than incorporate. You have used integrate in the rest of the document.

The bottom of page 4 and top of page 5, the two sentences that start 'The value of this study...' and 'Also unique to our study...' These sentences are about your choice of methodology and should be included in your methodology section, not in the background to the study.

Page 5, line 36 Participant should be participants.

Page 5, line 47. After research assistant put the abbreviation (RA) and then in the rest of the document replace research assistant with RA.

Page 6, line 31. For the purposes of this study and for readers to understand the rigour of your study you need to describe your experience/qualifications as the lead author in relation to mental illness.

Page 7: A little more details is needed with regards to your choice of certain sections of the MINI and not others. What was your justification in the use of this particular tool and only certain sections of it. Also can you justify why you administered the questionnaire in English? This is very surprising given that all participants home-language was Xhosa.

Ethics: More detail is required with regards to ethics. Did all participants sign informed consent? Was it administered in English? Were participants ensured of anonymity?

Discussion

Page 13, line 6. 'We found a higher rate of mental illness in patients with TB.' As you didn't have a comparison group, this is not a finding from your study, so you cannot say this. You can only refer to studies which have documented this.

Page 14, line 36: The paragraph that starts 'In this study, we did not find a statistically significant relationship...' is confusing and not logical. You need to clarify that the proportion of TB patients with mental illness is different in different studies. In line 42, is the use of the term 'prevalence' the correct term? Line 52 is a repeat of line 35.

Page 16, lines 15 – 19. The sentence that starts 'Alcohol use disorder...'. I suggest you delete this sentence as it is not related to your study.

In the conclusion, can you suggest a couple of areas of further research in this area?

	There are 77 References. Surely this is more than the journal allows? The number of references should be reduced.
--	---

VERSION 2 – AUTHOR RESPONSE

Reviewer

Thank you for asking me to review the revised version of this manuscript which describes the extent of mental health co-morbidity in TB patients in the Eastern Cape in South Africa. This is a well-written paper, however, I have a number of comments below which need to be addressed.

Response: We thank the reviewer for acknowledging the merits of our study.

Consistent use of terminology

I got quite confused reading the paper as the authors used various terms to refer to mental ill-health. These include: mental illness, mental morbidity, mental disorder, psychiatric condition, psychiatric disorder, mental illness, psychiatric comorbidity, comorbid psychiatric disorders, psychiatric diagnoses, mental disturbance. The authors need to decide on one term and use it consistently throughout the manuscript. In addition, this term should be defined in the methodology. If more than one term is needed as the terms describe different things (surely there can't be more than 2 or 3 terms needed at the most), these should all be defined in the methodology.

Response: We acknowledge the reviewer's comments. The terms mental disorder and psychiatric disorder are common and interchangeable words in literature, so we did not initially include their definition. The definition of this terminology is provided, and these will be used interchangeably throughout the text. Also, we think that psychiatric comorbidity or comorbid psychiatric disorder are commonly used and interchangeable, so we retain them but provided we have added the definition on the methodology.

Results section

It is unnecessary to repeat in the text all that is in the table. So, for example, from Table 1 only include what you think are the most important variables in the text. The age categories in Table 1 should be clarified – how old are adults or those who are middle-aged or aged+?

Response: We acknowledge the reviewer's comments and suggestions. We have indicated the age ranges for the different age categories in Table 1. The variables (relationship status, level of education, and employment status) chosen in the text from Table 1 are commonly associated with different mental disorders, so that is the reason for highlighting them in the text.

The results need to be reported in the accepted academic format. For example, when reporting different adherence in HIV-positive vs HIV-negative participants, the accepted format for reporting this is (77.8% HIV positive vs XX% HIV negative; $p=0.4$)

Response: We thank the reviewer for the suggestion, and we have changed the format as suggested. See page paragraph 2, page 11.

In the abstract, the last sentence in the objectives needs revising as it doesn't make sense: 'This study explored the psychiatric comorbidity and its clinical in individuals receiving TB treatment.'

Response: We thank the reviewer for pointing this out, and we have corrected the grammatical error.

Page 4, line 5: Use the term integrate rather than incorporate. You have used integrate in the rest of the document.

Response: We thank the reviewer, and we have changed the word from incorporation to integration (paragraph 3, page 4).

The bottom of page 4 and top of page 5, the two sentences that start 'The value of this study...' and 'Also unique to our study...' These sentences are about your choice of methodology and should be included in your methodology section, not in the background to the study.

Response: We thank the reviewer for the comment. Previous reviewers had asked us to include in the introduction the uniqueness of our methodology in this study because there are many previous studies evaluating the prevalence of mental disorders in individuals with tuberculosis. So, after review, we have retained the two statements because we think they will be helpful to the reader.

Page 5, line 36 Participant should be participants.

Response: We appreciate the reviewer for raising this point, and we have corrected the grammatical error.

Page 5, line 47. After research assistant put the abbreviation (RA), and then in the rest of the document replace research assistant with RA.

Response: We thank the reviewer for the suggestion, and we have used the abbreviation throughout page 6 as recommended.

Page 6, line 31. For the purposes of this study and for readers to understand the rigour of your study you need to describe your experience/qualifications as the lead author in relation to mental illness.

Response: We acknowledge the reviewer's suggestion, and we have added the lead author's qualification (see paragraph 2 on page 6).

Page 7: A little more details is needed with regards to your choice of certain sections of the MINI and not others. What was your justification in the use of this particular tool and only certain sections of it. Also can you justify why you administered the questionnaire in English? This is very surprising given that all participants home-language was Xhosa.

Ethics: More detail is required with regards to ethics. Did all participants sign informed consent? Was it administered in English? Were participants ensured of anonymity?

Response: We thank the reviewer for raising the above comments. We used all the MINI modules in our study and have since clarified that in the manuscript (last paragraph of page 7). The issue of language was raised by previous reviewers, and we responded as follows:

There is a translated Xhosa version that is not yet validated, and it has fewer translated diagnostic modules (less than half of the English version). However, we used the Xhosa MINI during the training of the research assistant (RA) to familiarize himself with the vocabulary as he was allowed to explain in Xhosa when a participant did not understand a statement. We acknowledge the potential limitation of administering an instrument in English to participants whose primary language is not English. Thus, we have included language as one of the study's limitations.

Informed consent was obtained from all study participants in their primary language of communication, and anonymity was ensured (see paragraph 1, page 6).

Discussion

Page 13, line 6. 'We found a higher rate of mental illness in patients with TB.' As you didn't have a comparison group, this is not a finding from your study, so you cannot say this. You can only refer to studies which have documented this.

Response: We acknowledge the reviewer for raising this point, and we have changed the wording from 'higher' to 'high' (first paragraph of page 13).

Page 14, line 36: The paragraph that starts 'In this study, we did not find a statistically significant relationship...' is confusing and not logical. You need to clarify that the proportion of TB patients with mental illness is different in different studies. In line 42, is the use of the term 'prevalence' the correct term? Line 52 is a repeat of line 35.

Response: We appreciate the reviewer for pointing out the lack of clarity in the highlighted statement. We have revised the statement as reflected in the last paragraph of page 14.

Page 16, lines 15 – 19. The sentence that starts 'Alcohol use disorder....'. I suggest you delete this sentence as it is not related to your study.

Response: We note the reviewer's comment, and after review, we have deleted the mentioned sentence.

In the conclusion, can you suggest a couple of areas of further research in this area?

Response: We thank the reviewer for this comment. There is evidence that treating comorbid psychiatric disorders in TB patients improves recovery. But there is a need for studies to evaluate whether simple management approaches such as brief interventions for substance use can improve adherence and prognosis in TB patients. In our setting, most patients with TB are managed at the primary care level, so such simple interventions would have a greater impact if proven to be effective for this population.

There are 77 References. Surely this is more than the journal allows? The number of references should be reduced.

Response: We thank the reviewer for making this comment. We were able to reduce the number of references after further review.